# Benchmarking Self-Supervised Vision Transformers in Astronomy

## Abstract

This work does not describe a novel method. Instead, it aims to extend the success of self-supervised pre-training on natural images to astronomical data. To address the lack of comprehensive benchmarks in astronomy, we first curate an unlabeled pre-training dataset and multiple datasets for typical astronomical tasks. Through extensive experiments, we demonstrate that our pre-training scheme has the following advantages. Representation transferability: pre-training followed by fine-tuning can not only significantly boost performance but also reduce training epochs compared to from-scratch training on downstream tasks (*e.g.*, improve 12% accuracy and reduce 83% epochs in galaxy classification), mirroring trends in natural image domains. Cross-instrument generalization: our pre-trained model generalizes across telescope instruments and outperforms domain models. Domain-specific pre-training data value: In-domain pre-training data further improves model performance and surpasses the results trained on general datasets such as ImageNet and other domain datasets. Furthermore, we explore Vision Transformers (ViTs) scaling in astronomy via parameter and data variation to offer insights and experiences for vision foundation model development in astronomy.

## 1 Introduction

The pre-training + transfer learning paradigm has revolutionized computer vision by significantly improving downstream tasks performance (Li et al., 2022; Fang et al., 2023) while substantially reducing computational and annotation costs (He et al., 2019; Dosovitskiy et al., 2020; He et al., 2022). Self-supervised learning methods, *e.g.*, contrastive learning (He et al., 2020; Chen et al., 2021; Chen & He, 2021; Grill et al., 2020) and masked image modeling (He et al., 2022; Chen et al., 2020; Bao et al., 2021; Xie et al., 2022; El-Nouby et al., 2024), eliminate manual annotation requirements, allowing pre-training on unprecedented scales (millions (Deng et al., 2009) to billions (Singh et al., 2023; Schuhmann et al., 2022) of images). However, despite its transformative impact on natural images, this paradigm remains largely unexplored in astronomy.

Astronomical data present two fundamental tensions that motivate our research. First, modern instruments generate petabyte-scale observational data (Dey et al., 2019; Mellier et al., 2024; Zhan, 2021), while curated annotations remain scarce. Second, intrinsic domain gaps – such as low signal-to-noise ratios (Sharma et al., 2020) and instrument-specific responses (Dey et al., 2019) – challenge direct transfer of natural image pre-training strategies. These disparities compel astronomers to spend years developing customized processing pipelines for individual instruments (Walmsley et al., 2020), underscoring the urgent need for foundation models with cross-instrument generalization capabilities.

As a first step, we examine three prerequisites for adapting self-supervised learning to astronomy:

(i) Pre-training method compatibility. Astronomical images share structural similarities with natural images, featuring three-band $(g, r, z)$ analogous to $(r, g, b)$ channels. This permits application of established pre-training methods that have been proven effective on natural images without obstacles. After pre-training, we fine-tuning the models to adapt to task with a specific format, *e.g.*, five-band $(u, g, r, i, z)$ inputs for photometric redshift estimation task.

(ii) Pre-training data curation. Current astronomy lacks systematically curated pre-training datasets. To our knowledge, AstroSSL (Stein et al., 2022), updated version of Hayat et al. (2021);

Stein et al. (2021), represents the *only one*[1] that provides 76 million unlabeled images (named `Astro-76M`) for self-supervised pre-training. However, the visual content in `Astro-76M` suffers from morphological homogeneity (Figure 1), which is not conducive to learning powerful representation (Fan et al., 2021; Fang et al., 2022; Huang et al., 2022). We address this issue by supplementing a subset with diversified image content (detailed description in Sec. 3).

(iii) Standard evaluation. Unlike natural images with standard benchmarks (*e.g.*, ImageNet-1K (Deng et al., 2009), COCO (Lin et al., 2014), and ADE20K (Zhou et al., 2017)), vision models for astronomy lack standard evaluation. Existing work employs ad hoc metrics (Hayat et al., 2021; Hausen & Robertson, 2020; Lin et al., 2021; Zhang et al., 2022; Lanusse et al., 2023), hindering comparative analysis. We build the first comprehensive benchmarks for typical astronomical tasks that can serve as standard pre-training testbed.

Having completed the aforementioned preparations, we focus on investigating three key questions:

- Transferability: Can the pre-training + transfer learning paradigm generalize to astronomy?
- Cross-instrument generalization: Do pre-trained models achieve cross-telescope adaptability?
- Domain-specific data value: Does natural image pre-training (*e.g.*, ImageNet (Deng et al., 2009)) benefit astronomical tasks? Does in-domain pre-training data further improve performance?

Our empirical results demonstrate the advantages of pre-training in astronomical contexts:

- Our pre-trained models achieve substantial performance enhancements over from scratch, *e.g.*, increasing morphology classification accuracy by 12% (Tables 2-4).
- Our pre-trained models demonstrate robust cross-instrument generalization capabilities, even outperforming strong domain models (Table 5).
- Domain-specific pre-training substantially exceeds natural image pre-training in efficacy, particularly for redshift estimation task: pre-training on astronomical data reduces prediction error by 64.6% than ImageNet pre-training (Table 6).
- Our pre-trained models show substantial improvements over existing ones, due to our enhanced pre-training dataset and efforts in astronomy-specific optimizations.

Our findings reveal that the pre-training + transfer learning paradigm's success in natural images is not merely replicable but amplifiable in scientific domains through curating pre-training data, designing comprehensive evaluation benchmarks and configuring model and training procedures.

## 2 RELATED WORKS

**Pre-training and Transfer Learning.** Modern vision tasks adopt a pre-training + transfer learning paradigm: a general-purpose, task-agnostic backbone is pre-trained through supervised (He et al., 2016; Dosovitskiy et al., 2020) or self-supervised learning (He et al., 2020; Chen et al., 2021; He et al., 2022), whose structure is later modified and adapted to the downstream tasks. Self-supervised pre-training, eliminates manual annotation requirements while enabling unprecedented scalability – leverage millions (Deng et al., 2009) even billions (Singh et al., 2023; Fan et al., 2025) of training samples, making it a powerful alternative to supervised training. The key advantage of pre-training is that it can not only significantly boost performance but also reduce training costs on downstream tasks. Our work investigates whether these established benefits can be effectively extended to astronomical data, where unique observational characteristics present both opportunities and challenges.

**Pre-training Data.** The efficacy of deep learning models fundamentally depends on training data quality and scale (Goyal et al., 2021; Chen et al., 2023). Within the pre-training paradigm, massive unlabeled data like ImageNet (Deng et al., 2009) and LAION-5B (Schuhmann et al., 2022) serve as foundational resources, enabling the development of vision foundation models through scalable self-supervised learning. To date, the astronomy community possesses only one self-supervised pre-training benchmark – the `Astro-76M` dataset (Stein et al., 2022), where focus on galaxies with accurate morphological classification by selecting targets with sufficient pixel numbers and low redshift values (Figure 1). Models pre-trained on such low-diversity dataset exhibit bias toward "clean" morphological features (Stein et al., 2022; Lanusse et al., 2023), hindering their ability to

---

[1] Angeloudi et al. (2024) provides a larger dataset, but neither releases pre-trained models nor conducts benchmark studies.

Table 1: **Summary of model architectures, pre-train data, and train recipes for C-MAE**. We employ models of varying scales and pre-training data sizes to investigate their scaling properties. We calculate the channel mean and standard deviation of the `DESI-2M` dataset for normalization, and search for the optimal hyperparameters, like base learning rate (blr) and weight decay (wd).

| encoder | decoder embed, depth, heads | | | parameters | pre-train data | data augmentation |
|---|---|---|---|---|---|---|
| ViT-B/L/H | 512, 1, 16 | | | 100/300/600M | `DESI-2M` | rand size+crop+flip |

| input size | mask ratio | optimizer | $(\beta_1, \beta_2)$ | batch size | wd | blr | epochs | precision | GPUs |
|---|---|---|---|---|---|---|---|---|---|
| 224×224 | 75% | AdamW | (0.9, 0.95) | 4096 | 0.05 | 2e-4 | 800 | fp16 | 64A100 |

process real-world observational data containing low signal-to-noise ratios objects. In this work, we build a new pre-training dataset by supplementing a subset with diversified image content. Previous finding (Fan et al., 2025) demonstrates that pre-training data distribution significantly impacts learned representations, underscoring the necessity of our augmented pre-training data.

**Downstream Task Benchmarking.** The computer vision community has established high-quality benchmark datasets, such as ImageNet-1K (Deng et al., 2009), COCO (Lin et al., 2014), and ADE20K (Zhou et al., 2017), to accurate evaluate model transfer capabilities. Astronomy, as a discovery-driven science, prioritizes scientific objectives over standardized methodological evaluations. For example, existing work (Stein et al., 2022; Hausen & Robertson, 2020; Lin et al., 2021; Zhang et al., 2022; Lanusse et al., 2023) predominantly employ ad hoc metrics tailored to individual instrument or object, precluding meaningful cross-study comparisons. To bridge this gap, our work establishes the first standard pre-training testbed and performs comprehensive evaluation for astronomical foundation models. We hope that these fundamental efforts on pre-training data and task-specific benchmarks can facilitate the development of artificial intelligence in astronomy.

## 3 PRE-TRAINING

Table 1 summarizes the pre-training settings, introduced next.

**Pre-training method**. In this work, we aim to extend the success of self-supervised pre-training on natural images to astronomical data, rather than designing a new method. Contrastive learning (Chen et al., 2021; Chen & He, 2021) will consume nearly 2× training resources, we consider using the more simple and efficient masked image modeling methods (He et al., 2022; Xie et al., 2022; Bao et al., 2021). Our pre-training follows the standard MAE (He et al., 2022) which takes the original image pixels as reconstruction target, as there was no image tokenizer available, prior to our work. During pre-training, we randomly mask a large subset (*e.g.*, 75%) of image patch and input the remaining visible patches to encoder. The encoded patches and mask tokens are processed by a lightweight decoder that reconstructs the original image in pixels. Building on the standard MAE framework and its application to data processing for the Chinese Space Station Telescope (CSST) (Zhan, 2021), we name our pre-training **C-MAE**.

**Architecture**. We use an asymmetric encoder-decoder architecture, similar to MAE. The encoder is a vanilla ViT (Dosovitskiy et al., 2020) with different parameters, to explore scaling behavior. The decoder adopts a lightweight design, which can further accelerate training.

**Pre-training data**. Contemporary astronomy is characterized by an abundance of observational data from large-scale sky surveys, *e.g.*, the Dark Energy Spectroscopic Instrument (DESI (Dey et al., 2019)) projects to accumulate 10 PB of data by the conclusion of 2025 and the CSST is expected to produce annual datasets exceeding 30 PB. However, current astronomy lacks systematically curated pre-training datasets. To the best of our knowledge, AstroSSL (Stein et al., 2022) currently represent the *sole one* in the literature that systematically curated 76 million unlabeled astronomical images (sampled from DESI, designated as `Astro-76M`) for self-supervised backbone pre-training via contrastive learning (He et al., 2020). This foundational effort has subsequently informed multiple research trajectories: LVM (Fu et al., 2024) adopted the `Astro-76M` corpus for autoencoder-based representation learning, while AstroCLIP (Lanusse et al., 2023) extended the paradigm through image *vs*. spectrum contrastive objectives with additional 0.2M pair data.

However, the `Astro-76M` dataset exhibits limited semantic complexity in its image content (Figure 1, row-1), where celestial objects primarily manifest low-variability morphological profiles. We posit that such structural homogeneity creates detrimental conditions for learning strong representations (Fan et al., 2021; Fang et al., 2022), as models tend to develop trivial solutions through low-level interpolation, even with a high mask ratio (Huang et al., 2022).

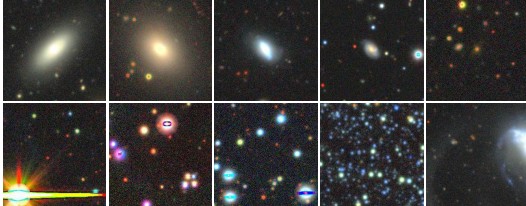

Figure 1: **Example training images**. The first row of images are from `Astro-76M`. The second row is from the 0.5M subset we added.

In this work, we conduct a new pre-train dataset that sample a subset (1.5M) from `Astro-76M`, along with supplementing a subset (0.5M) with more complex image content (row-2 in Figure 1). We refer to the 2M unlabeled data as `DESI-2M`. Table 3 quantifies the contribution of the 0.5M diverse samples. Further details on dataset construction are provided in the Appendix A. Moreover, we uniformly sample 1M and 0.5M data from `DESI-2M`, creating `DESI-1M` and `DESI-0.5M`, respectively. The consistent data distribution ensures fair comparisons of data scaling experiments. We employ `DESI-0.5M` to search for pre-training hyper-parameters, while using `DESI-1M` for default model pre-training under the optimized configuration.

## 4 EXPERIMENTS

We develop C-MAE variants with parameter counts of 100M, 300M, and 600M, to test the scaling capability. These variants follow standard ViTs architectures (Dosovitskiy et al., 2020) and scaling rules (El-Nouby et al., 2024). The detailed model configurations are shown in Appendix B. We conduct extensive experiments on typical astronomical tasks, and also evaluate models on more realistic and valuable cross-telescope settings. We will open source pre-train data, task-specific datasets, and our code to foster future research.

### 4.1 CAN THE PRE-TRAINING + TRANSFER LEARNING PARADIGM GENERALIZE TO ASTRONOMY?

Our insight is that for this paradigm to be valid, pre-training must satisfy two criteria: (1) achieving superior results with less training cost (*e.g.*, fewer training epochs or less annotated data) compared to trained from scratch; (2) pre-trained representations can be transferred to various downstream tasks. Next, we demonstrate that C-MAE has these capabilities.

#### 4.1.1 GALAXY MORPHOLOGY CLASSIFICATION

**Dataset.** We collect 20K galaxy images from DESI. These images share the same source telescope as our pre-training data, and we named `galaxy-desi`. Its eight categories are derived from a series of questions and answers organized by the famous Galaxy Zoo 2 project (Willett et al., 2013), *e.g.*, round elliptical, cigar-shaped elliptical, barred spiral. We split the training and test set into a 0.8/0.2 ratio. Moreover, we randomly sample subsets from the training set for few-shot experiments.

**Baseline.** We train classic ResNet (He et al., 2016) and ViT from scratch as baselines. Additionally, the baseline also includes three publicly available pre-trained models: LVM (Fu et al., 2024), AstroSSL (Stein et al., 2022), and AstroCLIP (Lanusse et al., 2023). All the three models were pre-trained on `Astro-76M` dataset. Differently, LVM adopts a symmetrical autoencoder framework, while AstroSSL and AstroCLIP use contrastive learning method. We search for learning rate, weight decay, drop path rate, and epochs, for each model size (18, 50, 101, B, L, H) and for each model type (ResNet, ViT, Swin). These hyper-parameters are included in the Appendix B.

**Results.** As detailed in Table 2, our C-MAE achieves significant improvement by 12% points compared to scratch models (*e.g.*, 87.23% *vs*. 74.74% from ResNet-101). This performance gain is attributed to pre-training, as only 50 epochs of fine-tuning were required (*vs*. 300 epochs from scratch). Other pre-trained models (LVM, AstroSSL, and AstroCLIP) also demonstrate higher accuracy and lower training costs than trained from scratch. This confirms the effectiveness of pre-training + transfer learning paradigm in astronomical data.

Table 2: **Galaxy morphology classification on** `galaxy-desi`. Our C-MAE achieves significant improvements in comparison to both scratch models and existing pre-trained models, achieving state-of-the-art results. AE: autoencoder, CL: contrastive learning, MIM: masked image modeling.

| method | backbone | pre-training | parameters | epochs | accuracy (%) |
|---|---|---|---|---|---|
| scratch | Res-18 | - | 11.18M | 300 | 71.64 |
| | Res-50 | - | 23.52M | 300 | 73.89 |
| | Res-101 | - | 42.52M | 300 | 74.74 |
| | ViT-Ti | - | 5.53M | 300 | 72.25 |
| | ViT-S | - | 21.67M | 300 | 73.94 |
| | ViT-B | - | 85.80M | 300 | 67.27 |
| | ViT-L | - | 303.31M | 300 | 73.02 |
| LVM | Swin-T | `Astro-76M`, AE | 56.53M | 50 | 84.63 |
| AstroSSL | Res-50 | `Astro-76M`, CL | 23.52M | 50 | 84.18 |
| AstroCLIP | ViT-L | `Astro-76M`, CL | 303.31M | 50 | 82.57 |
| **C-MAE** (ours) | ViT-B | `DESI-1M`, MIM | 85.80M | 50 | 87.23 |
| | ViT-L | `DESI-1M`, MIM | 303.31M | 50 | 88.38 |
| | ViT-H | `DESI-1M`, MIM | 681.27M | 50 | **89.10** |

Surprisingly, C-MAE outperforms models pre-trained on `Astro-76M` by 2.6%–4.5% accuracy using significantly less data. This large performance improvements can be explained by two factors. **(1)** Our augmented pre-training data. Recalling our analysis in Sec. 3, the images in `Astro-76M` exhibit low-variation morphological attributes. Such simplistic image content is not conducive to learning powerful visual representations. In contrast, our `DESI-1M` dataset contains richer and more diverse image data. The 3.0% accuracy gains in C (with diverse samples) versus B – with identical data volume and the base Astro-0.5M samples in Table 3 – directly quantifies the contribution of the 0.5M diverse samples, using ViT-B as encoder and C-MAE for pre-training in both cases.

**(2)** Our effort on astronomy-specific optimizations. We ablate key designs, details in Appendix C, and observe differences from natural images pre-training. We argue that these gaps are mainly due to the low signal-to-noise ratio of astronomical images. These redesigns significantly improve the learned representation, leading to 1.8% accuracy gains. In contrast, existing models (LVM, AstroSSL, and AstroCLIP) directly adopt the pre-training approaches developed for natural images.

Figure 2 shows few-shot results. Our C-MAE pre-trained ViT-B achieves 69.55% accuracy with only 800 labeled images, surpassing the best scratch-trained model (67.27% accuracy with 16,000 labeled images). This demonstrates C-MAE's capability to reduce the need for labeled data in downstream tasks. Furthermore, C-MAE consistently outperforms LVM across different training data sizes, and scaling the backbone leads to higher accuracy.

Table 3: **Quantitative contribution** of the 0.5M diverse samples using three different pre-training data. **A**: 0.5M images randomly sampled from `Astro-76M`; **B**: A + another 0.5M images from `Astro-76M`, for a total of 1M samples; **C**: A + the 0.5M diverse samples. C differs from the `DESI-1M` dataset, which can be roughly regarded as "Astro-0.75M + 0.25M diverse samples".

| pre-training data | A | B | C |
|---|---|---|---|
| accuracy (%) | 83.17 | 84.39 | 87.46 |

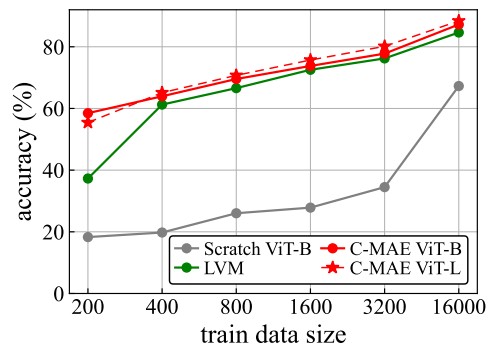

| train data size | 200 | 400 | 800 | 1600 | 3200 | 16000 |
|---|---|---|---|---|---|---|
| scratch, ViT-B | 18.29 | 19.97 | 26.02 | 27.84 | 34.49 | 67.27 |
| C-MAE, ViT-B | 58.47 | 63.95 | 69.55 | 73.78 | 77.78 | 87.23 |

Figure 2: **Few-shot morphology classification** on `galaxy-desi` test set. Our C-MAE pre-training achieves higher accuracy than trained from scratch while using less labeled data, *e.g.*, 800 *vs.* 16000.

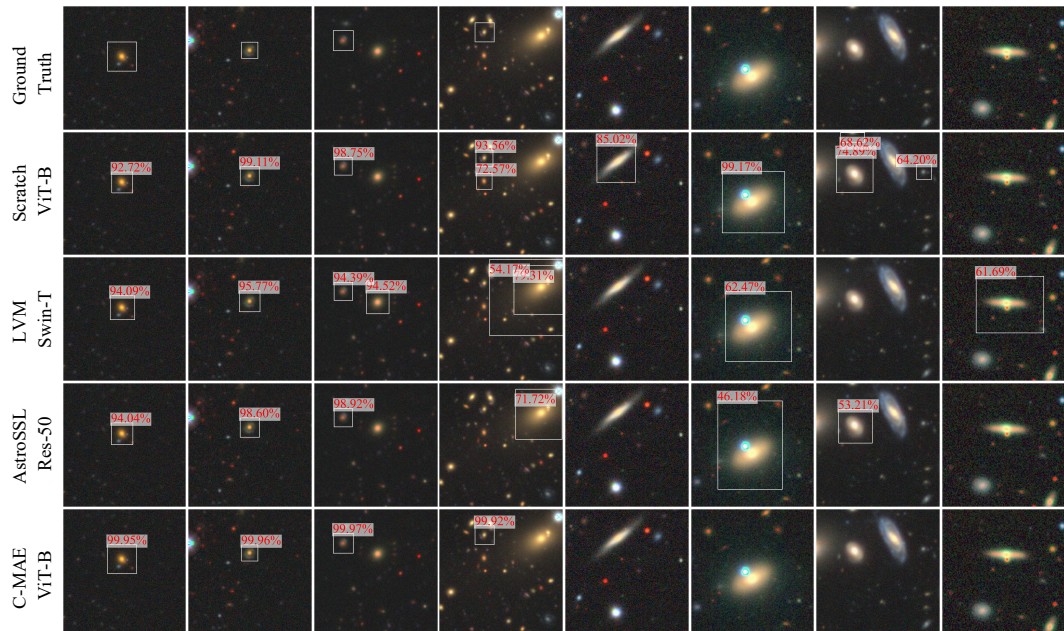

Figure 3: **Visualization results** on `neuralens-desi` test set. All baseline methods (2-4 rows) exhibit systematic output false positives results (3-8 columns), our C-MAE shows no such issue.

Table 4: **Lensing detection** on `neuralens-desi`. Pre-trained models outperform from-scratch using fewer training iterations, confirming the transferability to object detection. The visualized outputs are shown in Figure 3.

| method | backbone | iters. | AP | $AP^{50}$ | $AP^{75}$ | $AP^S$ | $AP^M$ | AR@1 | AR@10 |
|---|---|---|---|---|---|---|---|---|---|
| scratch | Res-50 | 64K | 21.29 | 45.21 | 16.87 | 24.01 | 25.06 | 37.34 | 57.65 |
| | ViT-B | 64K | 28.95 | 44.36 | 32.37 | 32.87 | 35.76 | 29.26 | 55.42 |
| LVM | Swin-T | 32K | 31.38 | 59.65 | 27.90 | 36.99 | 37.30 | 38.64 | 61.16 |
| AstroSSL | Res-50 | 32K | 31.76 | 58.37 | 29.95 | 31.69 | 37.56 | 39.58 | 60.44 |
| AstroCLIP | ViT-L | 32K | 27.53 | 54.84 | 23.49 | 31.14 | 33.25 | 34.94 | 56.38 |
| **C-MAE** (ours) | ViT-B | 32K | 39.88 | 49.22 | 45.70 | 38.84 | 81.74 | 41.83 | 42.54 |
| | ViT-L | 32K | 41.31 | 49.37 | 47.04 | 39.72 | 85.19 | 43.39 | 43.74 |
| | ViT-H | 32K | **42.62** | 51.46 | 49.43 | 41.97 | 87.64 | 45.10 | 45.68 |

### 4.1.2 STRONG GRAVITATIONAL LENSING DETECTION

Strong gravitational lenses, caused by foreground massive galaxies or clusters, distort the light from background sources. Their detection (Jia et al., 2022b; Li et al., 2024) is challenging due to their extremely low spatial density compared to ordinary galaxies. Consequently, detecting them requires an extremely low false-positive rate to be scientifically useful.

**Dataset.** Following LVM (Fu et al., 2024), we create a dataset from the `NeuraLens` database (Huang et al., 2021), providing center coordinates of candidate lensing systems. We generate 256×256 pixel cutouts (3-channel $g, r, z$ composite at 0.262 arcsec/pixel) centered on each candidate with a [-256, 256] pixel random offset for augmentation. Bounding-box annotations were created using OpenCV and Labelme[2] tools. The resulting dataset, named `neuralens-desi`, contains 11.6K images (6K that show strong lensing) and is divided into training and test sets with 88%/12% ratio.

**Implementation.** We use Detectron2 (Wu et al., 2019) with the influential Mask R-CNN framework (He et al., 2017). For ViT and Swin backbones, we follow ViTDet (Li et al., 2022), which is specially designed for vision transformer. We train ResNet and ViT/Swin backbones with SGD and AdamW (Loshchilov & Hutter, 2019), respectively. The input size is 512×512, augmented during training by large-scale jitter (Ghiasi et al., 2021) with a scale range of [0.1, 2.0]. All methods differ only in their backbone (type and pre-training), which ensures alignment of low-level details.

---

[2]https://opencv.org/, https://labelme.io/

Table 5: **Transferring pre-trained models to SDSS telescope**. Photo-Net is a strong baseline and significantly outperforms other pre-trained models. Our C-MAE outperforms the ensemble results. [†]: pre-trains backbone by contrastive learning; [‡]: combines 6 models.

| method | backbone | cross telescope | galaxy-sdss | | redshift-sdss | | | |
|---|---|---|---|---|---|---|---|---|
| | | | epochs | accuracy (%) | epochs | $\Delta z \downarrow$ | $\sigma_{\mathrm{MAD}}$ | $\eta$ (%) |
| scratch | Res-50 | ✗ | 100 | 85.46 | 50 | 16.31e-4 | 2.57e-2 | 0.92 |
| | ViT-B | ✗ | 100 | 87.38 | 50 | 4.32e-4 | 2.43e-2 | 0.83 |
| SC-Net[†] | 6conv+2fc | ✗ | 50 | 89.13 | - | - | - | - |
| Photo-Net[‡] | Inception | ✗ | - | - | - | 1.70e-4 | 1.43e-2 | 1.26 |
| LVM | Swin-T | ✓ | 50 | 93.48 | 50 | 13.82e-4 | 40.81e-2 | 0.48 |
| AstroSSL | Res-50 | ✓ | 50 | 95.26 | 50 | 7.37e-4 | 1.98e-2 | 1.11 |
| AstroCLIP | ViT-L | ✓ | 50 | 93.03 | 50 | 3.47e-4 | 2.38e-2 | 0.86 |
| **C-MAE** (ours) | ViT-B | ✓ | 50 | 96.00 | 50 | 0.97e-4 | 2.31e-2 | 1.09 |
| | ViT-L | ✓ | 50 | 95.76 | 50 | 0.77e-4 | 1.42e-2 | 1.57 |
| | ViT-H | ✓ | 90 | **96.02** | 85 | **0.51e-4** | 0.56e-2 | 2.43 |

**Results.** Table 4 reports results in COCO format (Lin et al., 2014), including AP (averaged over IoU thresholds), $AP^{50}$, $AP^{75}$, $AP^S$, $AP^M$ (AP at different scales), and AR@1, AR@10. Pre-trained models (except AstroCLIP) outperformed models trained from scratch, requiring fewer iterations (32K *vs.* 64K). Our C-MAE achieved the best performance, with a +10.9 point improvement in AP and +12.6 in AR@1 compared to scratch-trained ViT-B (37% and 43% relative improvement). Example outputs are visualized in Figure 3. While other baselines exhibit systematic false positives (3-8 cols in Figure 3), C-MAE did not. This is of great practical significance for the subsequent astronomical data analysis due to the extremely unbalance distribution of strong gravitational lensing.

## 4.2 DO PRE-TRAINED MODELS ACHIEVE CROSS-TELESCOPE ADAPTABILITY?

Cross-telescope data processing is vital for modern astronomical research, which involves integrating massive heterogeneous datasets from various observatories to support multi-wavelength and multi-messenger analysis. This study explores whether a pre-trained model on one telescope's data (data from DESI (Dey et al., 2019)) can perform well on another's (data from SDSS (York et al., 2000)), addressing a key issue in cross-device data processing.

**Dataset.** We evaluate models on two different tasks.

- The `galaxy-sdss` is a classification dataset (Zhang et al., 2022) derived from SDSS data release (Alam et al., 2015), comprising 5 classes, 23,037 training images, and 5,754 validation images. Validation accuracy is reported.
- The `redshift-sdss` was constructed using a sample of 12 major galaxies from the SDSS data release (Alam et al., 2015), accessible via the CasJobs interface [3]. After applying the SQL query from (Pasquet et al., 2019) and excluding objects with poor spectroscopic measurements ($Z_{\mathrm{WARNING}} \neq 0$), 41,650 galaxies were selected with $r$-band-dereddened Petrosian Magnitude $r < 17.77$ (survey completeness limit) and reliable spectroscopic redshifts ($0.01 < z < 0.3$). The input images are 224×224 pixels (0.396 arcsec/pixel) with 5-band ($u, g, r, i, z$), retrieved from the SDSS Science Archive server using the Astroquery package (Ginsburg et al., 2019). The dataset is divided into a training set (10,100 images) and a test set (31,550 images). Results are evaluated using common statistical metrics (Hogg et al., 2000): residuals $\Delta z$, median absolute deviation $\sigma_{\mathrm{MAD}}$, and outliers fraction $\eta$.

**Baseline.** We build two additional baselines that were trained on SDSS (without cross-telescope). Specifically, SC-Net (Zhang et al., 2022) pre-trains backbone on the `galaxy-sdss` training set by contrastive learning (Chen & He, 2021) and then conducts supervised classification; Photo-Net (Pasquet et al., 2019) is an ensemble model integrating six sub-models for redshift estimation. We replicate their reported results. For other baselines and our C-MAE, we attach a linear regressor to the final features of the backbone network for adaptation to the redshift estimation task. The loss function computes the mean squared error between the prediction and truth redshift.

---

[3] `https://skyserver.sdss.org/CasJobs/`

Table 6: **Compare pre-training strategies** with ViT-B backbone. Although ImageNet pre-training can help learn astronomical tasks, astronomical data pre-training can further improve performance. IN-1K and `DESI-1M` have similar amount of training data.

| | galaxy-desi | galaxy-sdss | neuralens-desi | | redshift-sdss | | |
|---|---|---|---|---|---|---|---|
| pre-train | accuracy | accuracy | AP | AR@1 | $\Delta z \downarrow$ | $\sigma_{\text{MAD}}$ | $\eta$ (%) |
| none (random init.) | 67.27 | 87.38 | 28.95 | 29.26 | 4.32e-4 | 2.43e-2 | 0.83 |
| IN-1K, supervised | 86.44 | 94.96 | 33.69 | 39.30 | 2.74e-4 | 0.75e-2 | 1.75 |
| IN-1K, MAE | 82.82 | 94.35 | 35.10 | 40.85 | 13.64e-4 | 1.93e-2 | 1.07 |
| `DESI-1M`, C-MAE | 87.23 | 96.00 | 39.88 | 41.83 | 0.97e-4 | 2.31e-2 | 1.09 |

**Results.** Table 5 presents the comparison results. For the `galaxy-sdss` classification (5-class, $\sim$23K images), all pre-trained models demonstrated significant performance improvements, with the weakest AstroCLIP outperforming SC-Net by 4% accuracy (93.03% *vs.* 89.13%).

However, for the more challenging redshift estimation task (limited training data), pre-trained models (LVM, AstroSSL, and AstroCLIP) performed markedly worse than Photo-Net and even underperformed trained from scratch (*e.g.*, 7.37e-4 *vs.* 4.32e-4). This highlights the difficulty in transferring across telescopes. Notably, our C-MAE (single-model result) achieved a residual of 0.97e-4, outperforming Photo-Net by 43% relative improvement, demonstrating that a well-designed pre-trained model can effectively generalize across different telescopes. Figure 4 illustrates the visualization of redshift predictions.

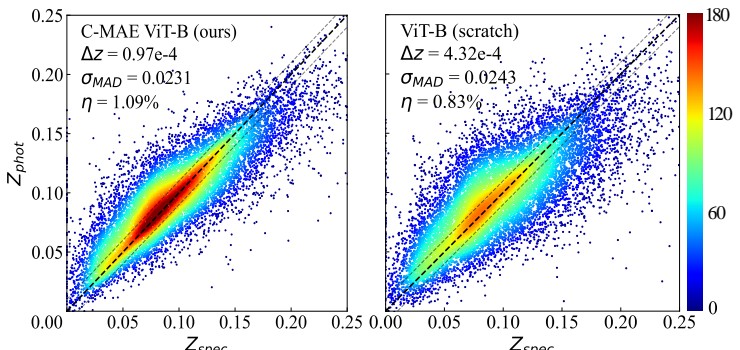

Figure 4: **Photo-z estimates on** `redshift-sdss` **test set**. Compare the fine-tuned C-MAE (left) to the equivalent scratch network (right).

### 4.3 DOES NATURAL IMAGE PRE-TRAINING BENEFIT ASTRONOMICAL TASKS?

The ImageNet-1K (IN-1K) dataset (Deng et al., 2009) is widely adopted in computer vision, owing to its potential to improve downstream task performance (Dosovitskiy et al., 2020; He et al., 2022; 2020). We transfer representations learned on IN-1K to astronomy tasks, as shown in Table 6, where evaluates both supervised and self-supervised models, the latter using MAE for pre-training.

Results demonstrate that supervised pre-training on IN-1K yields significant improvements over no-pretraining baseline, achieving gains of 19% on `galaxy-desi`, 4.7% AP on `neuralens-desi`, and reducing $\Delta z$ by 37% on `redshift-sdss`. In contrast, MAE pretraining exhibits mixed efficacy: while it surpasses the no-pretraining in lensing detection, it produces bad result on redshift estimation, far worse than the non-pretrained model.

We hypothesize that supervised pre-training fosters discriminative feature representations critical for cross-domain adaptation, whereas MAE's focus on reconstructing local image patches prioritizes spatially oriented features (Li et al., 2022; Park et al., 2023), which can benefit spatial tasks such as detection but are less transferable to regression-based tasks such as redshift estimation.

Notably, despite the DESI-1M dataset's similar scale to IN-1K (1M *vs.* 1.28M images), our C-MAE achieves superior performance in all tasks. This highlights that while natural image pre-training provides a foundational advantage, domain-specific pre-training on astronomical data offers substantial additional benefits. These findings underscore the interplay between pre-training objectives and downstream task requirements in domain shift scenarios.

We have included more results from natural image pre-training (DINO-v2 (Oquab et al., 2023) and CLIP (Radford et al., 2021)) in the Appendix C. Although these models utilized larger pre-training datasets or more sophisticated training procedures, they still underperform compared to our C-MAE, which further strengthens our conclusion.

Table 7: **Scaling pre-training data.** Increasing pre-training data from 0.5M to 2M, all models show better results, but tend to saturate. Both large scale image samples and sufficient training iterations are important for pre-training. The lower $\Delta z$ is better.

| pre-train data | epochs | accuracy on `galaxy-desi` | | | $\Delta z$ on `redshift-sdss` | | |
|---|---|---|---|---|---|---|---|
| | | ViT-B | ViT-L | ViT-H | ViT-B | ViT-L | ViT-H |
| *same pre-training epochs* | | | | | | | |
| `DESI-0.5M` | 800 | 85.95 | 87.05 | 87.43 | 1.72e-4 | 1.14e-4 | 1.29e-4 |
| `DESI-1M` | 800 | 87.23 | 88.38 | 89.10 | 0.97e-4 | 0.77e-4 | 0.51e-4 |
| `DESI-2M` | 800 | 87.65 | 88.41 | 89.24 | 1.13e-4 | 0.63e-4 | 0.33e-4 |
| `DESI-1M` | 1600 | 87.38 | 88.32 | 89.17 | 1.04e-4 | 0.76e-4 | 0.60e-4 |
| `DESI-2M` | 400 | 86.38 | 87.94 | 88.52 | 1.38e-4 | 0.89e-4 | 0.92e-4 |

## 4.4 SCALING PRE-TRAINING DATA.

We conduct ablation studies on key components of our pre-training framework (*e.g.*, decoder design, mask ratios, and training epochs), observing behaviors differ from MAE's image pre-training (He et al., 2022) (see Appendix C for details).

This section focuses on evaluating how scaling pre-training data impacts downstream performance. Our prior experiments primarily used 1 million images for pre-training. Table 7 now examines the effects of data scale on galaxy morphology classification and redshift estimation. While performance improves when increasing pre-training data from 0.5M to 2M, we observe diminishing returns (cf. `DESI-1M` *vs*. `DESI-2M` with 800 epochs). This aligns with prior findings (Singh et al., 2023; Xie et al., 2023) showing limited benefits of data scaling in vision models – for instance, MAE pre-trained ViT-L achieves only 1.0% improvement despite $14\times$ more pre-training data and 512px resolution (Singh et al., 2023), unlike the strong scaling law observed in language models (Brown et al., 2020; Kaplan et al., 2020). Developing vision learning methods that effectively utilize large-scale data remains an open challenge.

To further determine whether performance gains are due to more samples or more iterations, we keep the total number of seen samples constant across different data scales by adjusting training epochs. Extending `DESI-1M` training from 800 to 1,600 epochs shows modest gains but still underperformed `DESI-2M` trained for 800 epochs (same seen samples). This suggests that both data scale and sufficient training iterations are crucial for representation quality. The performance gap between 400-epoch `DESI-2M` and 800-epoch `DESI-1M` further confirms this.

## 5 DISCUSSION AND CONCLUSION

**Societal impacts.** Pre-trained models may inherit biases from the pre-training data and generate inexistent content. Our work does not sufficiently evaluate C-MAE on imbalanced data distributions, nor investigate whether and how they are biased as well.

**Limitations and future work.** First, we did not design a new self-supervised learning method but utilized existing ones. Future work could consider incorporating astronomical data characteristics (*e.g.*, equipment noise, target uncertainty) into the pre-training scheme; Second, during the curating of the pre-training data, we can further consider its characteristics to provide a more systematic data diversity control scheme; Finally, in addition to fine-tuning, we also explored linear probing and visual prompt tuning (Jia et al., 2022a; Wang et al., 2025), but these approaches did not achieve satisfactory results. Future work should focus on designing more efficient algorithms specifically tailored for astronomy to reduce the adaptation cost of foundation models.

**Conclusion.** This work extends self-supervised pre-training to astronomical data. We create benchmarks by curating an unlabeled pre-training dataset and task-specific datasets. Key findings include: Pre-training improves downstream task performance and reduces training costs; Pre-trained models generalize well across telescopes; In-domain pre-training data enhance model adaptation. We further explore the parameters and data scaling capacity. We hope that this work can provide useful insights and experiences for developing vision foundation models in astronomy.

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

## A  PRE-TRAIN DATASET

The dataset is derived from the January 2021 public release of Dark Energy Spectroscopic Instrument Legacy Imaging Surveys Data Release 9 (DESI DR9 [4]). Following the data processing pipeline described in `Astro-76M` (Stein et al., 2022), we first filter stellar objects from the Tractor Catalog [5] by selecting only sources whose best-fit morphological models deviate from the stellar point spread function.

From this preprocessed catalog, we initially extract 1.5 million galaxies, generating $3 \times 256 \times 256$ pixel cutouts (0.262 arcsec/pixel resolution) centered on each galaxy's equatorial coordinates using DESI DR9 JPEG imaging data. To augment the dataset, we supplement these with 0.5 million additional cutouts from background regions. Furthermore, we improve dataset quality by removing samples containing edge artifacts where blackened regions exceed 10% of the image area – a process demonstrated to reduce loss spikes (Molybog et al., 2023) and enhance training stability during model pre-training.

Pre-train data critically impact representation quality. The field of natural image is undergoing a shift in pre-train data: transitioning from the classic ImageNet (Deng et al., 2009) to the emerging MetaCLIP datasets (Fan et al., 2025; Bolya et al., 2025). While this work has primarily enhanced pre-train data through a straightforward implementation, future research could explore more sophisticated data curation strategies for higher-quality datasets, potentially even integrating multimodal inputs from multiple observational sources.

## B  IMPLEMENTATION DETAILS

**Architectures.** Our C-MAE adopts an asymmetric encoder-decoder architecture. The encoder uses the standard ViT (Dosovitskiy et al., 2020) architectures that have a stack of Transformer blocks (Vaswani et al., 2017). Each block consists of a multi-head self-attention layer and an MLP layer with LayerNorm (Ba et al., 2016). Refer to Table 8 for details about the encoder. The decoder adopts a lightweight design (ablation in Table 9) and is discarded after pre-training. Moreover, we do not use [CLS] token (He et al., 2022) during pre-training, and treat the global average pooling on the image tokens sequence as input for the task head.

**Hyper-parameters.** We search for the learning rate, weight decay, drop path rate, and epochs, for each model size (18, 50, 101, B, L, H), each model type (ResNet, ViT, Swin), and each downstream task. The hyper-parameters are included in Tables 10, 11 and 12.

Table 8: **Encoder architectures**. These variants adopt scale rules in El-Nouby et al. (2024).

| architecture | Layers | Patch size | Embedding dim | MLP size | Heads | Parameters |
|---|---|---|---|---|---|---|
| ViT-Base | 12 | 16 | 768 | 3,072 | 12 | 86M |
| ViT-Large | 24 | 16 | 1,024 | 4,096 | 16 | 303M |
| ViT-Huge | 24 | 14 | 1,536 | 6,144 | 16 | 680M |

## C  ADDITIONAL ABLATION AND RESULTS

We ablate some basic components related to C-MAE pre-training in Table 9 and Figure 5, and observe some phenomena different from those in natural images, introduced next. The default setting is 800-epoch pre-training C-MAE on `DESI-1M` with ViT-B as encoder, and reports fine-tuning and linear probing accuracy (%) on `galaxy-desi` test set.

**Decoder design.** The decoder consists of a set of Transformer blocks (Vaswani et al., 2017) and can be flexibly designed in a manner that independent of the encoder. Table 9 studies its depth (number of Transformer blocks) and width (embedding dim). C-MAE needs a shallow decoder (1∼4 blocks), both for linear probing and fine-tuning. Increasing the depth will produce very bad results; for example, when the depth is 12, the accuracy is reduced by half in linear probing. This

---

[4] `https://www.legacysurvey.org/dr9`
[5] `https://www.legacysurvey.org/dr9/description/#the-tractor-catalogs`

Table 9: **Ablation of decoder design.** We report top-1 accuracy (%) on `galaxy-desi` under the protocol of linear probing and fine-tuning. The default settings are marked by `grey`.

<table>
<tr><td colspan="3" align="center">(a) decoder depth.</td><td colspan="3" align="center">(b) decoder width.</td></tr>
<tr><td>depth</td><td>linear probing</td><td>fine tuning</td><td>width</td><td>linear probing</td><td>fine tuning</td></tr>
<tr><td>1</td><td>**53.21**</td><td>**87.23**</td><td>256</td><td>45.19</td><td>87.52</td></tr>
<tr><td>2</td><td>44.82</td><td>87.21</td><td>384</td><td>45.45</td><td>87.65</td></tr>
<tr><td>4</td><td>53.10</td><td>87.12</td><td>512</td><td>**53.21**</td><td>87.23</td></tr>
<tr><td>8</td><td>32.76</td><td>86.05</td><td>640</td><td>48.61</td><td>**87.72**</td></tr>
<tr><td>12</td><td>27.27</td><td>85.42</td><td>768</td><td>47.23</td><td>87.69</td></tr>
</table>

Figure 5: **Mask ratio.** Left: 75% mask ratio works significantly better than others for both fine-tuning (top) and linear probing (bottom). The y-axes are accuracy on `galaxy-desi` test set. Right: example reconstruction results on `galaxy-desi` test images. The C-MAE is pre-trained with a mask ratio of 75% but applied on inputs with other mask ratios. Although different from the ground truth, it is semantically reasonable.

phenomenon is different from observations in natural images, which MAE (He et al., 2022) works with a sufficiently deep decoder. We infer that this difference is caused by the different content complexity of astronomical images and natural images: compared to natural images, astronomical images have simpler contents, where celestial objects manifest low-variability morphological profiles, so it does not require a deep decoder to reconstruct the missing contents. Table 9b varies decoder width. We use 512-d by default, which performs well under linear probing and fine-tuning.

**Mask ratio.** Figure 5 left shows the influence of the mask ratio. Surprisingly, linear probing and fine-tuning exhibit similar trends with varying mask ratios. The optimal mask ratio, 75%, works significantly better than other values. This is different from the observation of MAE in natural images (ImageNet-1K), where a wide range of mask ratios (40∼80%) work well for fine-tuning. Figure 5 right shows some reconstruction results using 75% mask ratio during pre-training.

**Training epochs.** Figure 6 varies the training epochs. The fine-tuning accuracy improves steadily with more training epochs, which aligns with MAE in natural image. From another perspective, the fact that more training epochs can give further improvement means that the model converges more slowly. We assume that this is due to the encoder only sees 25% of patches per

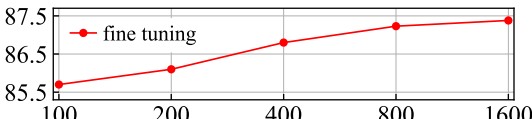

Figure 6: **Training epochs**. Each point is a full training schedule. The y-axis is accuracy on `galaxy-desi` test set.

image, which speeds up training but also loses some meaningful image content.

**Some ineffective tricks.** Beyond standard MAE (He et al., 2022), we also tried some other tricks during pre-training, including using a fixed random patch projection (Chen et al., 2021) layer to embed the image patches, learnable position embedding (Dosovitskiy et al., 2020; Chen et al., 2021), LayerScale (Touvron et al., 2021; 2022), and converting MaskFeat (Wei et al., 2022) into a regularization loss, but did not observe any benefit on performance or training stability.

**Astronomy-specific optimizations** involve two aspects: (1) We enhanced the pre-training data by supplementing it with a richer set of images, and during pre-training, we removed training samples containing artifacts. This process has been demonstrated to reduce loss spikes (Molybog et al., 2023)

and enhance training stability. (2) We redesigned the decoder in an astronomical context. Compared to the optimal configuration for natural images, our redesign achieved a 1.8% accuracy gain.

**More results of natural image pre-training.** We evaluate DINO-v2 (Oquab et al., 2023) and CLIP (Radford et al., 2021) on morphological classification and redshift estimation tasks and show results to the table below, using ViT-B as encoder in all cases. Note that the publicly available DINO-v2 model was pre-trained on LVD-142M, and CLIP on LAION-2B.

| pre-train | IN-1K, sup | IN-1K, MAE | LVD-142M, DINO-v2 | LAION-2B, CLIP | DESI-1M, C-MAE |
|---|---|---|---|---|---|
| accuracy (%) | 86.44 | 82.82 | 87.15 | 86.78 | 87.23 |
| $\Delta z \downarrow$ | 2.74e-4 | 13.64e-4 | 3.59e-4 | 8.25e-4 | 0.97e-4 |

These larger scale natural image models ultimately underperform our DESI-1M. This performance gap demonstrates the necessity of pre-training on astronomical data for astronomical tasks.

## D CODE AND DEMO

**Code.** We include our self-contained codebase (refer to the zip file `Code-CMAE`) as a part of the supplementary material. Please refer to `README.md` for instructions how to use the code. We do not include model weights in the supplementary material as they are too large (>100MB) that exceed the space limit. We will open source pre-train dataset, task datasets, and our code to foster research.

**License**. We release open-source code under the MIT license to foster future research in this field.

**Requirement**. Running our `Python` code requires some common packages, such as PyTorch, TorchVision, and timm. Please refer to `Code-CMAE/README.md` for more details.

**Demo.** We use Jupyter Notebook to create three demos, including evaluating image classification and detection results and displaying the redshift prediction. See `demo-accuracy-eval.ipynb`, `demo-detection-eval.ipynb`, and `demo-redshift-eval.ipynb` for more details.

Table 10: Fine tuning C-MAE on different tasks. Multiple values in a cell are for different models.

| config | galaxy-desi | neuralens-desi | galaxy-sdss | redshift-sdss |
|---|---|---|---|---|
| optimizer | AdamW | AdamW | AdamW | AdamW |
| momentum | $\beta_1, \beta_2 = 0.9, 0.999$ | $\beta_1, \beta_2 = 0.9, 0.999$ | $\beta_1, \beta_2 = 0.9, 0.999$ | $\beta_1, \beta_2 = 0.9, 0.999$ |
| weight decay | 0.05, 0.5, 0.5 | 0.1 | 0.3, 0.1, 0.05 | 0.5, 0.1, 0.5 |
| batch size | 64, 64, 32 | 64 | 64, 64, 32 | 64, 64, 32 |
| learning rate | 1.5e-3, 2e-3, 1e-3 | 5e-4 | 1.5e-3, 2e-3, 1e-3 | 1.5e-3, 1e-3, 1.5e-3 |
| lr schedule | cosine decay | multi step | cosine decay | cosine decay |
| layer-wise lr decay | 0.65, 0.75, 0.75 | 0.7, 0.9, 0.9 | 0.65, 0.75, 0.65 | 0.65, 0.65, 0.75 |
| training epcohs | 50 | 32K iters | 50, 50, 90 | 50, 50, 85 |
| warmup epochs | 5 | 500 iters | 5 | 5, 5, 10 |
| drop path | 0.1 | 0.1 | 0.1 | 0.1 |
| EMA | 0.9999 | - | 0.9999 | 0.9999 |

Table 11: Training ResNet and ViT from scratch on galaxy-desi.

| config | ResNet-18/50/101 | ViT-Ti/S/B/L |
|---|---|---|
| optimizer | SGD | AdamW |
| optimizer momentum | 0.9, w/ Nesterov | $\beta_1, \beta_2 = 0.9, 0.999$ |
| weight decay | 5e-3 | 5e-2 (Ti/S), 1e-2 (B/L) |
| batch size | 128 | 128 |
| learning rate | 1e-3 | 1e-3 (Ti/S), 1e-4 (B/L) |
| learning rate schedule | multi step, [160, 230, 290] | cosine decay |
| training epcohs | 300 | 300 |
| warmup epochs | 0 | 10 |

Table 12: Fine tuning LVM, AstroSSL, and AstroCLIP on galaxy-desi.

| config | LVM
Swin-T | AstroSSL
ResNet-50 | AstroCLIP
ViT-L |
|---|---|---|---|
| optimizer | AdamW | SGD | AdamW |
| optimizer momentum | $\beta_1, \beta_2 = 0.9, 0.999$ | 0.9, w/ Nesterov | $\beta_1, \beta_2 = 0.9, 0.999$ |
| weight decay | 1e-3 | 1e-3 | 1e-3 |
| batch size | 64 | 128 | 64 |
| learning rate | 1e-4 | 1e-2 | 1e-4 |
| learning rate schedule | cosine decay | multi step, [30, 45] | cosine decay |
| layer-wise lr decay | 0.65 | - | 0.75 |
| training epcohs | 50 | 50 | 50 |
| warmup epochs | 5 | 5 | 5 |
| drop path | 0.1 | - | 0.1 |
| EMA | 0.9999 | 0.9999 | 0.9999 |

