# OpenReview forum: "Benchmarking Self-Supervised Vision Transformers in Astronomy"
_ICLR.cc/2026/Conference — Submitted to ICLR 2026_

### Official Review · Reviewer_PzGD · 2025-10-31

**Soundness:** 3
**Presentation:** 3
**Contribution:** 2
**Rating:** 6
**Confidence:** 3

**Summary:**

This paper studies the effectiveness of self-supervised training, in particular, the Masked Auto-Encoder (MAE) framework, on astronomical data.

The paper points out several limitations of existing works on the same topics such as lack of diversity in training data, lack of astronomy-specific optimization in the training pipeline, and ad-hoc evaluations and metrics. The paper makes several contributions to address these issues:
- Building a more diverse dataset consisting of a 1.5M subset of the existing Astro-76M dataset and a 0.5M set of more complex content.
- Optimize the training pipeline with optimized designs for astronomical data.
- Standardizing benchmarks by introducing `galaxy-desi` dataset for galaxy morphology classification and `neuralens-desi` dataset for lensing detection.

The paper shows that self-supervised pre-training is effective for astronomical data. increasing significantly the performance on benchmarks compared to the baselines that are trained from scratch. It also shows that in-domain pre-training brings more benefits than  pre-training on natural images, and the resulted model, C-MAE, generalizes well to cross-telescope data.

**Strengths:**

Although self-supervised learning (SSL) has been shown effective on many cases, its benefits are not clear on visual domains that are far from natural images. The paper's effort to confirm the benefits of SSL on astronomical data is significant and could potentially encourage more research in this area.

The paper is in general well written and organized, despite some inclarities that will be specified in "Weaknesses". Many experimental results are provided to support different claims in the paper. In the supplemental, the paper also lists some design choices that are ineffective which, I think, as as important as showing choices that work.

I appreciate the paper's focus on data curation and the importance of pre-training data to the model's performance. Indeed, the pre-training data introduced in the paper seems to bring substantial gains compared to the existing Astro-76M dataset. Finally, the paper also highlights astronomy-specific design choices in the model which bring further performance gain.

**Weaknesses:**

1. Given that the introduction of the new pre-training and evaluation data is probably the most important contributiion of the paper, I think that  the space dedicated to describing the different datasets is somewhat limited. In particular:
- The paper mentions that the 1.5M dataset is a subset of Astro-76M but did not discuss whether it is sampled uniformly from Astro-76M or there is a change in distribution of the images compared to Astro-76M. In their paper, Stein et al. provided subtaintial details about the data pipeline. I expect the paper to include a description with a similar level of details.
- The paper adds a 0.5M dataset of "background" images  but does not specify they are obtained from which galaxies and whether galaxies figured in this 0.5M set overlap with those depicted in the 1.5M set.
- Similarly, I wonder if there is any overlap between the galaxy-desi dataset and the two datasets mentioned above.
- The paper introduces 2 new benchmarks `galaxy-desi` and `neuralens-desi` but little information about their quality is provided. In particular, is there any control on the correctness of the labels. On `neuralens-desi`, obtaining bounding boxes with opencv and labelme
is quite adhoc.

2. Though interesting, the observations in the paper are hardly surprising.  The effectiveness of SSL on non-natural images have been observed in medical images, satellite images, etc. The fact that SSL pre-training helps reducing the number of labelled data  is well-known.

3. There are some missing ablation studies such as comparing DESI-2M to a dataset consisting of only `background` images, a breakdown on the contribution of the pre-training data and the astronomy-specific optimization.

**Questions:**

1. Could the authors provide more information about the datasets as specified in `Weaknesses`?
2. Could the authors provide additional ablation studies mentioned in `Weaknesses`?
3. Could the authors provide more insight as to why introducing 0.5M `background` images improves the results? The explanation about data diversity is a bit vague. I wonder if the 0.5M set correlates well with the benchmarks? Could adding even more `backgroung` images further help? Is the 1.5M part necessary at all?
4. I also like to know the authors' opinions on how to further pushing the use of SSL pre-training in astronomical data.

---

> ### Author Response · Authors · 2025-11-20
>
> **Response-Part I**
>
> We sincerely thank Reviewer PzGD for the positive assessment of our work. The reviewer describes our paper as "well written and organized", notes that it provides "many experiments to support claims", and states that it "confirms the benefits of SSL on astronomical data and could potentially encourage more research in this area". We are particularly grateful for the reviewer's recognition of our "focus on data curation and the importance of pre-training data". We address the major concerns below.
>
> >**Reviewer PzGD requires more information regarding the relationship between the pre-train DESI-2M dataset and the existing Astro-76M dataset, as well as details about the annotation quality of the two new benchmarks: galaxy-desi and neuralens-desi.**
>
> - **The relationship between DESI-2M and Astro-76M dataset.**
>
>     We randomly sampled 1.5M images following the data processing pipeline described in Astro-76M. This sampling strategy can be **approximately regarded as uniform**. We then supplemented the subset with an additional 0.5M diverse samples. The resulting combined collection of 2M unlabeled data constitutes the DESI-2M dataset.
>
>     It is important to note that the supplemental 0.5M samples also originate from the DESI survey but are **not** part of the Astro-76M dataset. This is because these 0.5M samples were not generated as cutouts centered on specific galaxies (the primary observational targets in Astro-76M), but rather from background regions -- areas of the sky that do not contain prominent astronomical targets. This deliberate inclusion enhances the diversity of the pre-training data.
>
> - **Annotation quality of galaxy-desi benchmark.**
>
>     The galaxy-desi dataset was created by generating additional cutouts following the same pipeline as Astro-76M. Consequently, there may be some overlap between galaxy-desi, the original Astro-76M dataset, and the 1.5M subset used in our pre-training data.
>
>     The eight classification categories for galaxy-desi are derived from the question-and-answer schema organized by the Galaxy Zoo 2 (GZ2) project. Recognizing that the raw GZ2 data contains many low-quality or ambiguous labels, we implemented a quality control procedure:
>     - We applied a threshold of 0.7 to the vote distributions, meaning only responses that achieved a vote fraction greater than 0.7 were assigned as the categories for a sample.
>
>     - Subsequently, the entire dataset underwent **manual verification** to ensure final annotation quality.
>
>
> - **Annotation quality of neuralens-desi benchmark.**
>
>     For the neuralens-desi benchmark, target regions of size 256x256 pixels were cutouted using OpenCV. The bounding boxes for these lensing candidates were then annotated **manually** using the Labelme tool, ensuring precise ground-truth labels for this detection task.

---

> ### Author Response · Authors · 2025-11-20
>
> **Response-Part II**
>
> >**Reviewer PzGD comments "Though interesting, the observations in the paper are hardly surprising". For instance, the effectiveness of SSL on non-natural images has been observed in medical images, satellite images, etc, and SSL pre-training helps reduce the number of labelled data is well-known.**
>
> While we agree that the effectiveness of self-supervised learning (SSL) on non-natural images has been demonstrated in domains like medical and satellite imagery, and that SSL's ability to reduce reliance on labeled data is well-established, our work moves beyond these established observations to reveal **two novel and significant phenomena specific to astronomical foundation models**:
>
> - **Cross-instrument generalization**. We demonstrate for the first time that a foundation model exhibits strong cross-instrument generalization -- it can be applied to data from other telescopes and even outperforms domain-specific models pre-trained on the target instrument (`cf. Table 5`). This capability has the potential to transform how astronomers conduct their work (`cf. lines 41-45`).
>
> - **Domain-specific pre-training value**. Across all downstream tasks, all natural image pre-trained models, regardless of their pre-training methodology (Sup, MAE, DINO-v2, CLIP) or the scale of their pre-training data (1M to 2B), lag significantly behind our astronomically pre-trained C-MAE. This significant performance gap underscores the necessity of astronomical data pre-training for core astronomical tasks (`cf. Table 6 and appendix lines 867-875`).
>
> These findings provide concrete, quantitative evidence that establishes unique requirements and opportunities for fundamental astronomical models, going beyond general knowledge of SSL in natural or non-natural images.
>
>
>
> >**Reviewer PzGD asks for additional ablation studies to quantify the contribution of the 0.5M diverse samples and to provide a breakdown of the performance gains attributable to the pre-training data versus the astronomy-specific optimizations.**
>
> - **Quantitative contribution of the 0.5M diverse samples.**
>     We have quantitatively evaluated the impact of the 0.5M diverse samples in `Table 3`. To address the reviewer's query directly and for their convenience, we reproduce the results of Table 3 and **include the result of pre-training with only the 0.5M diverse samples** for comparison.
>
>     Experiments setting. We use ViT-B encoder and report fine-tuning accuracy on galaxy-desi with different pre-training datasets:
>
>     - (A) 0.5M images randomly sampled from Astro-76M,
>
>     - (B) A + another 0.5M images from Astro-76M, for a total of 1M samples,
>
>     - (C) A + the 0.5M diverse samples,
>
>     - (D) New! Only using the 0.5M diverse samples.
>
>     | pre-train data | A | B | C | D |
>     |:--:|:--:|:--:|:--:|:--:|
>     | accuracy | 83.17 | 84.39 | 87.46 | 81.25 |
>
>     The results indicate that incorporating the diverse samples into the pre-training data substantially enhances the representation quality (**C vs. B**). However, models trained exclusively on these 0.5M diverse samples perform poorly (**D vs. A**). We hypothesize that this inferior performance stems from the lack of sufficient astronomical morphological information in these diverse samples.
>
>
> - **Quantitative contribution of astronomy-specific optimizations**.
>     Our astronomy-specific optimizations involve two aspects (`cf. appendix lines 862-865`): (1) We enhanced the pre-training data, and removed training samples containing artifacts during pre-training; (2) We redesigned the decoder in astronomical context (For detailed analysis of why decoder is the primary factor, please refer to our response to Reviewer zwhx's fourth comment).
>
>     Here, we focus on quantifying the impact of the decoder redesign. To disentangle the individual contributions of *pre-training data* vs. *astronomy-specific optimizations*, we conducted the following controlled comparison:
>
>     - C-MAE (ours) Trained on the DESI-1M dataset using our astronomically-optimized decoder (1 transformer block);
>
>     - MAE (baseline) Trained on the identical DESI-1M dataset but using the original MAE configuration (8 transformer blocks) optimal for natural images.
>
>     | model | galaxy-desi, Acc | neuralens-desi, AP |  redshift-sdss, $\Delta z \downarrow $ |
>     |:-----:|:---------:|:--------:|:-----------:|
>     | C-MAE (ours) | 87.23 | 39.88 | 0.97e-4 |
>     | MAE (baseline) | 86.05 | 38.74 | 1.42e-4 |
>
>     The comparative results demonstrate that, **even when using exactly the same pre-training data, our astronomy-specific decoder optimization yields significant performance gains across downstream tasks.**

---

> ### Author Response · Authors · 2025-11-20
>
> **Response-Part III**
>
> >**Reviewer PzGD questions why introducing the 0.5M diverse samples improves the pre-train performance?**
>
> We believe the performance improvement from incorporating the 0.5M background samples into pre-training can be attributed to the following three factors:
>
> - **Increased training data diversity**. If the pre-training data consisted solely of centered, simple-morphology astronomical objects, the model might rely on simplistic local cues (e.g., bright galactic cores) for reconstruction without understanding the global morphology of the sources. The inclusion of background region data disrupts this *simple pattern*, compelling the model to learn more global and fundamental image features to complete the pre-training task. This process encourages the learning of more generalizable representations.
>
> - **Serving as negative samples.** Incorporating background data as *negative samples* during pre-training significantly enhances the model's ability to distinguish astronomical signals from complex background noise. For instance, in tasks like strong gravitational lensing detection, this leads to a lower false positive rate (`cf. Fig. 3`).
>
> - **Noise estimation.** The background in astronomical images is not merely simple Gaussian noise but comprises complex structures including instrumental noise, sky background fluctuations, and cosmic ray residuals. Performing masking and reconstruction on background data enables the model to learn the statistical properties of such noise. When downstream data contains similar noise patterns, the model can better resist interference, resulting in more robust predictions.
>
> Regarding whether adding more background images would further improve model performance, we believe this requires rigorous experimental validation. For example, one could hold the total pre-training data volume constant and systematically adjust the proportion of background images within the range [0, 1]. Conducting this series of experiments, however, necessitates substantial computational resources and time, which, unfortunately, we cannot currently support due to the prohibitively high cost. We aim to investigate this question in our future work.
>
>
> >**Reviewer PzGD inquires about how to further advance the use of SSL pre-training with astronomical data.**
>
> We believe progress can be made in two key directions:
>
> - **Pre-training methodology**. Future work could explore incorporating astronomical data characteristics -- such as instrumental noise and target uncertainty -- into the pre-training objective. This would enable the model to predict not only pixel values but also noise levels and associated uncertainties, leading to more robust and calibrated representations.
>
> - **Pre-training data**. Another promising direction is the fusion of multi-telescope data during pre-training. By creating a unified dataset covering larger sky areas from diverse instruments, we can encourage the model to learn a unified representation of the sky and develop more powerful cross-instrument generalization capabilities.

---

### Official Review · Reviewer_zwhx · 2025-10-31

**Soundness:** 3
**Presentation:** 3
**Contribution:** 2
**Rating:** 4
**Confidence:** 5

**Summary:**

This paper aims to adapt and benchmark self-supervised ViTs for astronomical imagery and establish the first large-scale foundation model and evaluation framework for this domain. They curate an unlabeled dataset of 2M images for pre-training, called DESI-2M, that is filtered from the Astro-76M dataset. The authors leverage this data to develop a pre-training strategy for masked auto-encoders, referred to as C-MAE. They perform fine-tuning and evaluation of C-MAE on three tasks: strong gravitational lensing detection (a localization task), galaxy classification, and redshift estimation (a regression task). The proposed C-MAE consistently improves performance on all of these tasks compared to from scratch fine-tuning and fine-tuning from ImageNet pre-trained models.

**Strengths:**

The problem is interesting and under explored. To the reviewer, who is not an expert in astronomy, a foundation model in this domain could be impactful given the large amount of objects in space that need to be studied. The massive amounts of unlabeled data coming from space instruments, such as telescopes, motivates the use of self-supervised pre-training

The paper proposes a comprehensive benchmark with standard datasets and evaluation protocols for pre-training and transfer in astronomy

The authors perform thorough ablations on their proposed pre-training strategy and demonstrate its effectiveness across three tasks

The paper is well written and the ideas are conveyed clearly

**Weaknesses:**

The primary weakness lies in the limited novelty of the work for a representation learning conference. Most of the experiments are identical to ones performed in the MAE paper [1], and the primary insight from this work is that domain-specific self-supervised pre-training is more effective than natural image pre-training in the astronomy domain. This insight is consistent with many past works [2, 3, 4] exploring self-supervised pre-training for novel domains
* The novelty does not have to come from the method, it can even come from the data filtering (e.g., [5]). For example, the authors mention limitations of existing large-scale astronomy datasets on Lines 162-171. Is it possible to quantify these limitations and show that they negatively affect the pre-training? Or that they are not necessary for pre-training?

As mentioned above, the authors mention limitations of existing large-scale astronomy datasets but do not quantify these limitations or explain them in detail. It is unclear how the proposed data filtering strategy mitigates these limitations in Astro-76M
* It is also unclear if the proposed data filtering is helping, as no experiment is done with C-MAE trained on Astro-76M (or a random subset to match the size of DESI-2M)

The paper mentions astronomy-specific optimizations of the MAE model, but it is unclear why the proposed optimizations are astronomy specific and not dataset specific as they are only evaluated empirically using the proposed pre-training data and benchmarks. The exploration of why these optimizations help in the astronomy domain is shallow and should be expanded

Some technical details were unclear to the reviewer. On Lines 48-52, it is mentioned that models are trained on three-channel inputs and fine-tuned on five-channel inputs, but how is the architecture adjusted to process these additional channels? The standard MAE uses a convolutional layer to “patchize” each image into tokens, where the convolution kernel weights are fixed for three-channel inputs. The kernel dimensions would no longer match in the fine-tuning setting where there are five-channel inputs.

Minor formatting comments:
* The headers in the tables should be bolded
* Figure 2 contains both a Figure and a Table

[1] He et al., Masked Autoencoders Are Scalable Vision Learners, CVPR 2022

[2] Azizi et al., Big Self-Supervised Models Advance Medical Image Classification, ICCV 2021

[3] Reed et al., Self-Supervised Pretraining Improves Self-Supervised Pretraining, WACV 2022

[4] Kang et al., Benchmarking Self-Supervised Learning on Diverse Pathology Datasets, CVPR 2023

[5] Suorong Yang et al., A CLIP-Powered Framework for Robust and Generalizable Data Selection. ICLR 2025

**Questions:**

A (potentially naive) question: what is the practical impact of a foundational astronomy model in the astronomy community? Highlighting this in the introduction would be helpful to convey the potential impact of this work

(The questions below correspond to the weaknesses listed above)
What new insight or contribution does this work provide beyond demonstrating that domain-specific pre-training outperforms natural-image pre-training?

Can the authors quantitatively demonstrate how the limitations of existing astronomy datasets (e.g., Astro-76M) affect pre-training performance?

Can the authors show direct experimental evidence that their data filtering strategy improves representation quality or downstream results?

What makes the proposed model modifications astronomy-specific rather than general dataset tuning choices?

How is the MAE patch-embedding layer modified to handle five-channel inputs during fine-tuning?

---

> ### Author Response · Authors · 2025-11-20
>
> **Response-Part I**
>
> Reviewer zwhx provides a positive assessment of our work, praising it as a paper that "is well written and the ideas are conveyed clearly", one that tackles an "interesting and under explored problem", and one that is supported by "thorough ablations" which "demonstrate its effectiveness across three tasks". We thank the reviewer for this supportive feedback and address the major concerns below.
>
> >**Reviewer zwhx asks "what is the practical impact of a foundational astronomy model in the astronomy community?" and suggests "highlighting this in the introduction".**
>
> Thank you for raising this profound question regarding the broader impact of foundation models for astronomy. We will incorporate a discussion of these points into the introduction of the final version.
>
> Astronomical foundation models, exemplified by our C-MAE framework, aims to catalyze a paradigm shift in astronomical data analysis. We believe its practical impact manifests across three key dimensions:
>
> - **Transition from fragmented models to a unified foundation**.
>
>     - *Consolidated Processing Pipelines*. A typical astronomical data processing pipeline comprises multiple sequential modules, including denoising, cosmic-ray removal, segmentation, detection, and image enhancement, each traditionally requiring separate model architectures. By adapting a single foundation model across these tasks through fine-tuning or prompt tuning, we transcend the conventional *one-model-per-module* approach, significantly streamlining deployment and system integration.
>
>     - *Cross-Instrument Generalization*. Instrument-specific variations in noise characteristics, resolution, and filter systems have historically necessitated developing specialized models for each new telescope or survey. Foundation models overcome this *one-model-per-instrument* limitation by maintaining robust performance across diverse observational systems. As demonstrated by C-MAE's cross-instrument generalization capability (`Table 5`), our model adapts effectively to new equipment with minimal fine-tuning, eliminating redundant development efforts and enabling astronomers to focus on scientific discovery rather than model engineering.
>
> - **Enabling new discoveries through enhanced generalization**. Trained on extensive and diverse datasets, astronomical foundation models learn fundamental representations of celestial phenomena. This enables superior performance on complex tasks where natural image pre-trained models prove inadequate. For instance, C-MAE substantially outperforms natural image models on astrophysically critical tasks including redshift estimation and strong gravitational lensing detection (`Table 6`). This demonstrates that domain-specific foundation models are essential for reliable astrophysical inference, promising more accurate scientific outcomes.
>
> - **Democratizing advanced data analysis**. Publicly available astronomical foundation models serve as powerful, readily deployable tools that provide astronomers -- including those with limited computational resources -- access to state-of-the-art processing capabilities. This accessibility accelerates research progress while lowering barriers to cutting-edge scientific investigation.
>
> In conclusion, astronomical foundation models represent a transformative advancement that moves the field beyond fragmented, task-specific solutions toward a unified analytical framework. Our C-MAE implementation provides compelling validation of this approach and its potential to reshape astronomical data analysis.

---

> ### Author Response · Authors · 2025-11-20
>
> **Response-Part II**
>
> >**Reviewer zwhx asks "What new insight or contribution does this work provide beyond demonstrating that domain-specific pre-training outperforms natural-image pre-training?"**
>
> Beyond this insight and the general consensus that pre-training significantly improves downstream performance, our work has another two main contributions:
>
> - **Establishing a standardized benchmark for astronomical foundation models**.
>
>     As you noted, *"The paper proposes a comprehensive benchmark with standard datasets and evaluation protocols for pre-training and transfer in astronomy".* We fully agree with this assessment.
>
>     We would like to clarify that the development and application of computer vision in astronomy currently lag behind those in natural images. We attribute this gap to the absence of standardized benchmarks in astronomy, including curated pre-training data and annotated datasets for downstream tasks. For instance, existing studies often rely on ad-hoc training datasets and evaluation metrics tailored to specific instruments or targets, which hinders cross-study comparisons (`cf. line 172`).
>
>     Therefore, in this work, we first rigorously benchmark pre-training data and diverse downstream task evaluation. These fundamental efforts help bridge critical gaps between astronomical and natural images in pre-training and evaluation, thereby facilitating broader development and application of AI in astronomy.
>
> - **Cross-Instrument Generalization.** We demonstrate for the first time that a foundation model, C-MAE, exhibits strong cross-instrument generalization -- it can be applied to data from other telescopes and even outperforms domain-specific models pre-trained on the target instrument (`cf. Table 5`). This capability has the potential to transform how astronomers conduct their work (`cf. lines 41-45, as also detailed in our response to the first question`).
>
>
> >**Reviewer zwhx requests a quantitative demonstration of the impact of the Astro-76M dataset on pre-training performance, as well as direct experimental evidence showing that the proposed data filtering strategy enhances representation quality.**
>
> We have in fact quantitatively compared the impact of a subset of Astro-76M versus our augmented DESI-2M dataset on pre-training performance (`Table 3`). For the reviewer's convenience, we reproduce the results of Table 3 and describe the experimental settings below.
>
> We would like to first clarify the relationship between DESI-2M and the existing Astro-76M dataset (`cf. lines 172-174`). We randomly sampled a 1.5M subset from the Astro-76M and supplemented it with an additional 0.5M diverse samples. We refer to this combined 2M collection of unlabeled data as DESI-2M. Note that the 0.5M supplemental samples also originate from DESI but are not part of Astro-76M. **This means our approach constitutes data augmentation rather than data filtering.**
>
> To directly quantify the contribution of the specially curated 0.5M diverse samples, we conduct an ablation study comparing three pre-training datasets:
>
> - (A) 0.5M images randomly sampled from Astro-76M,
>
> - (B) A + another 0.5M images from Astro-76M, for a total of 1M samples,
>
> - (C) A + the 0.5M diverse samples.
>
> All other pre-training parameters remained identical. We use ViT-B encoder and report fine-tuning accuracy on the morphological classification task below:
>
> | pre-train data | A | B | C |
> |:--:|:--:|:--:|:--:|
> | accuracy | 83.17 | 84.39 | 87.46 |
>
> The 3.0% accuracy gains in C (with diverse samples) versus B -- with identical data volume and the base Astro-0.5M samples -- directly quantifies the contribution of the diverse samples.

---

> ### Author Response · Authors · 2025-11-20
>
> **Response-Part III**
>
> >**Reviewer zwhx questions:"What makes the proposed model modifications astronomy-specific rather than general dataset tuning choices?"**
>
> We thank the reviewer for raising this insightful question, which helps clarify the core motivation behind our optimizations. We agree that empirical performance improvements alone are insufficient to demonstrate their *astronomy-specific* nature. Our optimizations -- such as the decoder design and masking ratio -- are indeed grounded in the fundamental physical characteristics of astronomical images, rather than being mere dataset-specific tuning choices. Below we elaborate on the underlying rationale.
>
> - **Optimizations stem from inherent physical properties of astronomical data, not statistical characteristics**
>
>     Fundamental differences between astronomical and natural images render pre-training strategies that excel on natural images suboptimal for astronomy. Our optimizations directly address these inherent distinctions:
>
>     - Low signal-to-noise ratio and simple morphology.
>         - **Astronomical characteristics.** Unlike content-rich, textured natural images, many astronomical sources appear as simple, low-dimensional morphologies (e.g., ellipses, disks) with low signal-to-noise ratios. Excessively high masking ratios risk discarding already scarce key signals, preventing meaningful reconstruction and learning. Conversely, overly low masking ratios may allow the model to infer targets via simple extending of lines and textures without comprehending the holistic structure of the target and scene. Thus, unlike in natural images where MAE operates effectively across a wide range of masking ratios (e.g., 40%-80%), astronomical images may exhibit optimal performance at a specific ratio.
>         - **Our optimization.** Consequently, we explored and identified an optimal masking ratio of 75%, which works significantly better than other values (`cf. appendix Fig.5`). This represents a design choice tailored to the *information sparsity* characteristic.
>
>     - Difference in reconstruction objectives.
>         - **Astronomical characteristics.** The core of astronomical analysis involves measuring physical parameters of celestial objects (e.g., brightness, shape, spectrum), rather than perceiving rich textures and edges. An overly complex, deep decoder (e.g., the 8-layer transformer decoder used in the original MAE for natural images) is prone to overfitting the simple morphologies in astronomical images. It is therefore unnecessary and potentially detrimental.
>         - **Our optimization.** We adopted a significantly shallower decoder design comprising only 1 layer (`cf. appendix Table 9`). This forces the encoder to learn and compress more robust, physics-relevant representations, while the decoder handles only the simple mapping from high-level representations to pixels. This enhances the quality of the encoder's output representations and improves training efficiency.
>
> - **Generalization across multiple tasks and instruments demonstrates *astronomy-specific* nature**
>
>     If our optimizations were merely *dataset-specific*, their effectiveness would be limited to other astronomical tasks or telescopes. However, as demonstrated in our paper:
>     - The C-MAE model, using the same configuration, achieves significant and consistent performance improvements across diverse downstream tasks, including **morphological classification, redshift estimation, and strong gravitational lensing detection.**
>
>     - More importantly, C-MAE exhibits strong cross-instrument generalization (`cf. Table 5`), indicating that **its learned representations are not dependent on the characteristics of data from any specific telescope but instead capture fundamental, instrument-agnostic astronomical structures.**
>
> The success across different tasks and data sources strongly suggests that our optimizations target the intrinsic properties of astronomical data, rather than being dataset-specific tuning choices.
>
> >**Reviewer zwhx asks:"How is the MAE patch-embedding layer modified to handle five-channel inputs during fine-tuning?"**
>
> The pre-trained C-MAE model uses a three-channel convolutional layer for patch embedding.  When adapting to downstream tasks with different input channels, such as the five-band (u, g, r, i, z) imagery used in photometric redshift estimation, we replace this pre-trained three-channel layer with a new, randomly initialized five-channel convolutional layer. Both this new layer and the rest of the network are then fine-tuned on the target task data.

---

> ### Comment · Reviewer_zwhx · 2025-11-27
>
> Thank you to the authors for their response. I am now convinced of the importance of a foundational model for astronomy, but still have issues with the overall contribution of the work. **In summary, I believe the scope of the paper is too broad, and no single aspect of SSL in astronomy was explored in enough depth to yield a significant knowledge gain to the community.**
>
> I will provide additional details here. The high level idea (taken from abstract) is to “extend the success of self-supervised pre-training on natural images to astronomical data”. To this end a few contributions are claimed: 1) the introduction of a new benchmark, 2) the curation of a new in-domain pre-training dataset, 3) an astronomy specific MAE, 4) the observation that C-MAE generalizes across instruments. These are all well motivated elements to explore, but the problem of this paper is that none of them were explored in sufficient depth.
> * The new benchmark: It is clear that the work proposes a testbed for evaluating models on astronomy tasks, but the creation of these benchmark (e.g., pipeline, statistics, diversity, annotation trustworthiness) are not discussed in enough detail to constitute this as the primary contribution
> * Curation of a new in-domain pre-training dataset: It is clear that the proposed pre-training dataset improves performance empirically, but the methodology to curate this dataset is a simple sampling from Astro76M and DESI
> * Astronomy specific MAE (C-MAE): From the paper, this seems to be the primary contribution that the authors aim to present. However, C-MAE at the implementation level is nearly identical to MAE, just with a lower masking ratio and shallow decoder, limiting its novelty. These two design choices are attributed to “low signal-to-noise ratio and simple morphology” and “difference in reconstruction objectives”, but neither of these properties were quantified, and no proof was given that the improvements from the chosen design choices successfully mitigate/improve these two properties of astronomy data, it is only shown that these choices improve performance on the chosen benchmarks. The experiments conducted here only seem to suggest the optimal MAE hyper parameters for astronomy, with no deeper justification
> * Generalization across instruments: This is an interesting observation, but again is not explored in enough detail to constitute a meaningful advancement, it is only empirically shown that C-MAE can perform this generalization and not why it enables this property
>
> To conclude, the paper focuses too much on breadth rather than depth, resulting in none of the corresponding elements being explored in enough detail to yield a significant knowledge gain to the community. I have a similar opinion to reviewer `PeUj`, who notes that the paper mainly shows “what works rather than why it works”. However, I realize the importance of foundational astronomy models and the fact that this topic is under explored, so will leave my rating as a 4 as this work can help motivate future works in astronomical foundation models.

---

### Official Review · Reviewer_7FpM · 2025-11-01

**Soundness:** 3
**Presentation:** 4
**Contribution:** 4
**Rating:** 6
**Confidence:** 4

**Summary:**

Presents a dataset and self-supervised pre-training strategy for learning strong representations for typical astronomical data tasks. The proposed data and approach is shown to lead to better performance, faster convergence, cross-instrument generalization. The paper also conducts comprehensive experiments on around scaling and in-domain pretraining for astronomy tasks.

**Strengths:**

– The paper studies a challenging, interesting, and  under-studied problem – of learning representations for astronomical data – and does so in a principled way. Its experiments are well-designed and likely to be of value to researchers in the community. The paper also includes code to reproduce its findings.

– The paper is extremely well-written, and clearly lays out each claim and its supporting evidence.

– The paper includes a comprehensive set of experiments, across model backbones, tasks (morphology classification, lensing detection, cross-telescope transfer), settings (few-shot v/s full finetuning), and data/parameter scaling

**Weaknesses:**

– The paper’s study would be more complete with the inclusion of pretraining strategies other than C-MAE to ascertain which SSL strategies work better/worse for this domain. Have the authors attempted benchmarking more lightweight contrastive learning approaches like SimSiam [A]?

[A] Chen et al., Exploring Simple Siamese Representation Learning, CVPR 2021

**Questions:**

Kindle address the weakness listed above.

---

> ### Author Response · Authors · 2025-11-20
>
> Reviewer 7FpM provides a positive assessment of our work, highlighting that the paper is "extremely well-written", tackles a "challenging, interesting, and under-studied problem... in a principled way", and features "well-designed" experiments of value to the community. We thank the reviewer for this supportive feedback and address the major concerns below.
>
> >**Reviewer 7FpM comments "the paper would be more complete with the inclusion of other pretraining strategies", and requests benchmarking results for the SimSiam method.**
>
> In this work, we select MAE over contrastive learning for four key reasons:
>
> - **Performance**. MAE outperforms typical contrastive learning-based methods, as supported by relevant benchmarks in the original MAE paper (Table 3). Although more advanced methods like DINO-v2 show marginally stronger results than MAE, they involve a significantly more complicated training process (cf. DINO-v2, Table 1).
>
> - **Efficiency**. MAE offers significantly greater computational efficiency during pre-training.
>
> - **Implementation**. Contrastive methods rely heavily on empirically validated techniques like memory banks, large batch sizes, momentum encoders, MLP projections, and fixed random patch projections. These components lack established efficacy on astronomical data. MAE does not require these tricks. Critically, our own attempts to incorporate several of these techniques during pre-training proved ineffective (Appendix, lines 857-861).
>
> - **Task generalization**. While contrastive learning excels at linear probing and shape-based classification (e.g., morphology),, its performance degrades on detection and regression tasks [R1, R2] -- such as strong gravitational lensing detection and redshift estimation in astronomy. MAE-style pre-training, on the other hand, demonstrates strong generalization across a broader range of astronomical tasks.
>
> [R1] What do self-supervised vision transformers learn?
>
> [R2] Exploring plain vision transformer backbones for object detection.
>
> In response to the reviewer's suggestion, we have added SimSiam results to the table below. Note that SimSiam was pre-trained on a 1M subset randomly sampled from the Astro-76M and our augmented DESI-1M datasets, respectively. We used a ResNet-50 encoder and trained for 300 epochs, following the official SimSiam implementation in all other aspects.
>
>
> | Pre-train | galaxy-desi, Acc | galaxy-sdss, Acc | neuralens-desi, AP | redshift-sdss, $\Delta z \downarrow $ |
> |:-----:|:---------:|:--------:|:-----------:|:-----------:|
> | Astro-1M, SimSiam, Res-50 | 84.01 | 94.83 | 31.54 | 8.72e-4 |
> | DESI-1M,  SimSiam, Res-50 | 86.17 | 95.66 | 34.07 | 3.13e-4 |
> | DESI-1M,  C-MAE,   ViT-B  | 87.23 | 96.00 | 39.88 | 0.97e-4 |
>
>
> The results show that SimSiam underperforms compared to C-MAE, particularly on detection and regression tasks. This aligns with findings in [R1], which suggest that contrastive learning-based methods primarily capture global relationships and tend to develop homogenized self-attention patterns.
>
> Furthermore, models pre-trained on our augmented DESI-1M dataset achieve significantly stronger performance than those trained on the baseline Astro-1M dataset (see first two rows). This reaffirms that the quality of the pre-training data plays a crucial role in learning powerful representations.
>
> We thank Reviewer 7FpM for the professional suggestion and will incorporate these results into the final version.

---

> > ### Comment · Reviewer_7FpM · 2025-11-25
> > **Thank you for the response**
> >
> > I appreciate the author response including the new experiments with SimSiam. I continue to think that this is a well-conducted study and likely to be of value to the vision+astronomy community, and will recommend acceptance.

---

### Official Review · Reviewer_PeUj · 2025-11-03

**Soundness:** 2
**Presentation:** 1
**Contribution:** 1
**Rating:** 2
**Confidence:** 3

**Summary:**

This paper considers training models for astronomical data. It does not claim to introduce a new method. Instead, there is a mix of claimed contributions:
- a new pre-train dataset that sample a subset from the existing Astro-76M  dataset
- some benchmark
- some pretraining scheme called C-MAE (mostly inspired by MAE)

**Strengths:**

The paper aims at addressing a need for standardization and comparison in deep learning astronomy by curating novel and diverse benchmark datasets.

**Weaknesses:**

In the first place, I was confused by the positioning of the paper. What is claimed to be to contribution is not very clear overall? The curation of the dataset? The benchmarking itself? The MAE-style variant for training? Depending on when we look in the paper, we have different. claims. Overall, it sounds a bit like a mix of all, with most of the paper dedicated to claiming superior performance for C-MAE over baselines that are questionable (See below).

The paper is largely focused on what works rather than why it works. There is a lack of depth in analyzing why certain SSP methods (e.g., specific masking strategies in MAE or contrastive loss variants like SimCLR) succeed or fail in the astronomical context compared to natural images.

In multiple recent papers, it was shown that Dino-v2 (And now v3) is the best model for transferring to new kind of data (biological, satellite, etc)., if the model is large enough. The paper only shows a small table in Appendix with the Vit-B model, which is clearly not where Dino is best at. I have multiple concerns:
* I see no good reason for choosing Vit-B for Dino-v2, especially since the paper considers Vit-L and Vit-H. Dino would be MUCH better.
* One can still fine-tuning a dino pretrained model, which in my humble opinion would evidence that the proposed pretraining scheme has no interest.

**Questions:**

please address the weaknesses.

---

> ### Author Response · Authors · 2025-11-20
>
> **Response-Part I**
>
> We appreciate the time and effort that reviewer PeUj has dedicated to providing a professional assessment of this work. Below, we address the main concerns raised.
>
> >**Reviewer PeUj comments "the contributions of this paper are somewhat unclear"**.
>
> First, we would like to clarify that the development and application of computer vision technologies in astronomy currently lag behind those in the field of natural images. We attribute this gap to the **absence of standardized benchmarks in astronomy -- including pre-training data and annotated datasets for downstream tasks**. For instance, existing studies often rely on ad-hoc training datasets and evaluation metrics tailored to specific instruments or targets, which hinders cross-study comparisons (`cf. line 172`).
>
> Therefore, in this work, we first rigorously benchmark three key components (or foundational preparations) of deep learning models: pre-training data, pre-training methods, and diverse downstream task evaluation. **This addresses critical gaps between astronomical and natural images in pre-training and comprehensive evaluation**.
>
> After completing these three foundational steps, we developed C-MAE, a pre-trained model for astronomical data, and evaluated its performance against other models on the established downstream tasks. Beyond the commonly observed benefit of pre-training in improving downstream performance, **C-MAE demonstrates unique and impactful capabilities not seen in previous models**:
>
> - **Cross-instrument generalization**. C-MAE can be applied to data from other telescopes and even outperforms domain-specific models pre-trained on the target telescope (`cf. Table 5`). This capability has the potential to transform how astronomers conduct their work (`cf. lines 41-45`).
>
> - **Domain-specific pre-training value**. Across all downstream tasks, all natural image pre-trained models, regardless of their pre-training methodology (Sup, MAE, DINO-v2, CLIP) or the scale of their pre-training data (1M to 2B), lag significantly behind our astronomically pre-trained C-MAE. This significant performance gap underscores the necessity of astronomical data pre-training for core astronomical tasks (`cf. Table 6 and appendix lines 867-875`).
>
> In summary, the contributions of this work are multifaceted. We hope that these fundamental efforts -- on pre-training data, task-specific benchmarks, and the C-MAE model -- will facilitate the development and application of artificial intelligence in astronomy.

---

> ### Author Response · Authors · 2025-11-20
>
> **Response-Part II**
>
> >**Reviewer PeUj comments "the paper is largely focused on what works rather than why it works"**.
>
> We respectfully note that this may be a misunderstanding. In fact, we provide in-depth analysis of key components, such as:
>
> - **Pre-training Data**. We analyze the limitations of the Astro-76M dataset and their impact on pre-trained representations (`cf. lines 162-171`), propose targeted solutions, and quantitatively measure the contribution of the 0.5M samples in `Table 3`.
>
> - **Model Architecture**. Through an analysis of astronomical data characteristics, we identified decoder design and masking ratios as key factors influencing pre-training performance (as detailed in our response to Reviewer zwhx's fourth comment). Our ablation study confirmed phenomena distinct from those observed in natural images, and these redesigns were demonstrated to significantly enhance model performance (`cf. Table 9 and Fig. 5`).
>
> >**Reviewer PeUj asks why we used ViT-B rather than the larger DINO-v2 model?**
>
> We respectfully clarify that the core objective of comparing astronomical pre-training with natural image pre-training is to investigate whether domain-specific pre-training offers meaningful value. If fine-tuning general foundational models pre-trained on natural images could yield satisfactory results, domain-specific pre-training would be unnecessary.
>
> Our experimental results demonstrate that all natural image pre-trained models lag significantly behind our astronomically pre-trained approach. It is important to note that all models in these comparisons use a ViT-Base encoder for a fair comparison.
>
> We acknowledge that using a larger model could potentially alter this outcome.
>
> To address the reviewer's query, we have now included ViT-Large results on both morphological classification (galaxy-desi) and redshift estimation (redshift-sdss) in the table below.
>
> - On the morphological classification task, DINO-v2 (ViT-L) achieves slightly better accuracy than C-MAE. However, it is critical to note that DINO-v2 was pre-trained on 142M images -- over 140 times more data than the 1M astronomical images used for C-MAE.
> Furthermore, the morphological classification task itself is relatively straightforward (shape-based classification), which particularly benefits methods like DINO-v2 that inherently learn strong discriminative features during pre-training.
>
> - In contrast, on the more complex and domain-specific redshift estimation task, DINO-v2 still lags significantly behind our astronomically pre-trained C-MAE, reinforcing the unique value of domain-specific pre-training for core astronomical challenges.
>
> | Pre-train | DINO-V2, ViT-B | DINO-V2, ViT-L | C-MAE, ViT-B | C-MAE, ViT-L |
> |:-----:|:---------:|:--------:|:-----------:|:-----------:|
> | galaxy-desi, accuracy | 87.15 | 88.41 | 87.23 | 88.38 |
> | redshift-sdss, $\Delta z \downarrow $ | 3.59e-4 | 2.83e-4 | 0.97e-4 | 0.77e-4 |

---

> ### Author Response · Authors · 2025-11-20
>
> **Response-Part III**
>
> >**Reviewer PeUj comments that fine-tuning a foundational vision model, such as one from the DINO family, is possible and therefore pretraining a domain-specific model from scratch is not meaningful**.
>
> Good question! We completely agree that domain-adaptive fine-tuning of a powerful general-purpose vision model like DINO is a highly efficient and viable strategy. This approach effectively leverages the rich visual representations (e.g., edges, textures, shapes) already learned by the general model and adapts them to new scientific data distributions at a relatively low cost. In many scenarios with limited data or computational resources, this is undoubtedly the preferred approach.
>
> However, we posit that for specific scientific domains like astronomical imaging, **the fundamental statistical properties of the data and the core concepts to be learned differ radically from those in natural images**. Specifically, astronomical images often exhibit the following characteristics:
>
> - Extremely low signal-to-noise ratios. Target signals are often drowned in noise, unlike the clear object boundaries in natural images.
>
> - Point sources versus extended sources. Stars are point sources, while galaxies are extended sources, which differs from the concept of 'objects' in natural images.
>
> - Multi-wavelength nature. Data often comes from different electromagnetic bands (X-ray, infrared, radio), requiring unique methods for information fusion.
>
> - Absence of a 'background' concept. Nearly every pixel in an astronomical image may contain valuable astrophysical information.
>
> **A model pretrained from scratch on vast amounts of unlabeled astronomical data has the opportunity to learn these intrinsic, foundational representations of the domain from the ground up, rather than merely correcting biases from natural images**. It might learn to better *ignore* instrumental noise, distinguish point sources from cosmic ray traces, and understand the physical correlations between different wavelengths -- fundamental capabilities that a general model might never fully master through fine-tuning alone.
>
> From a longer-term perspective, pretraining a foundational model from scratch in a scientific field like astronomy holds significance that transcends performance on any specific downstream task (e.g., classification, detection). The goal is to construct a 'visual foundational world model' for the field. This specialized model holds the potential to:
>
> - Become a powerful, shared feature extractor for the entire astronomy community.
>
> - Discover novel, previously unlabeled patterns within the data, thereby driving hypothesis-free scientific discovery.
>
> - Provide a base that truly *understands* astronomical data for processing future large-scale sky survey data (e.g., from CSST).
>
> While fine-tuning DINO can also serve downstream tasks, achieving this depth of domain understanding and discovery potential likely requires the more fundamental path of pretraining from scratch.
>
> As mentioned in our response to the previous question, we compared our model (C-MAE), pretrained from scratch, against fine-tuned DINO models. On the photometric redshift estimation task, which requires deep understanding, C-MAE demonstrated a significant advantage. This preliminarily validates the potential of domain-specific pretraining for capturing the essential features of astronomical images.
>
> In summary, we do not dismiss the practical value of fine-tuning. Instead, we wish to emphasize that *pretraining from scratch* holds unique and strategic significance for building deeply domain-specific scientific foundation models. These two pathways serve different objectives:
>
> - *Fine-tuning general models* is an efficient engineering solution for specific scientific tasks.
>
> - *Domain-specific pretraining* is a deep scientific exploration into the intrinsic structure of a field, aimed at constructing its foundational model.
>
> Our work is dedicated to exploring the latter path. We believe that building a *native* visual model for astronomy from the ground up will deliver long-term value to the field's foundational model ecosystem.

---

### Author Response · Authors · 2025-12-02
**A summary note at the end of author-reviewer discussion period**

Dear PCs, SACs,  ACs and Reviewers,

Thank you for overseeing the reviewing process. The active author-reviewer discussions make the openreview threads go too long for you to quickly grasp the state of our rebuttal. Therefore, we drop this message to summarize our paper, the reviews, and our rebuttal.

**Summary of our paper.**
Currently, the progress in computer vision for astronomy significantly lags behind that for natural images. We attribute this gap to the **absence of standardized benchmarks in astronomy, including pre-training data and annotated datasets for downstream tasks**.

Therefore, in this work, we first rigorously benchmark three key components of deep models: pre-train data, pre-train methods, and diverse downstream task evaluation. **This addresses critical gaps between astronomical and natural images in pre-training and comprehensive evaluation**.

Upon completing these foundational steps, we analyzed the distinctions between astronomical and natural images. For instance, astronomical sources typically exhibit simple, low-dimensional morphologies (eg, ellipses, disks), coupled with low signal-to-noise ratios. These inherent data differences indicate that model architectures proven effective on natural images cannot be directly applied. We identified the decoder and mask ratios as critical factors influencing the quality of pre-train representations. Our redesigning for these components leads to significant improvements (`cf. appendix`). Further, we developed C-MAE models tailored for astronomical data, and compared its against existing models. Beyond the well-documented benefit of pre-training for downstream task enhancement, **C-MAE exhibits unique and impactful capabilities not observed in prior models**:

- **Cross-instrument generalization**. C-MAE can be applied to data from other telescopes and outperforms domain-specific models pre-trained on the target telescope (`cf. Table 5`).

- **Domain-specific pre-training value**. Across all downstream tasks, all natural image pre-trained models, regardless of their pre-training methods (Sup, MAE, DINO-v2, CLIP) or data scale (1M to 2B), lag significantly behind our C-MAE. This underscores the necessity of astronomical data pre-training for core astronomical tasks (`cf. Table 6`).

We hope that these fundamental efforts on pre-training data, task-specific benchmarks, and the C-MAE models, will facilitate the development of AI in astronomy.

**Summary of reviews and rebuttal.**
Reviewers provided highly positive feedback on our work. They commended the paper as "extremely well-written" for tackling a "challenging, interesting, and under-studied problem" in a principled manner. The experiments were praised as "well-designed" and considered valuable to the community. Reviewers also highlighted that the study successfully "confirms the benefits of SSL on astronomical data", which could encourage further research in this area. Particular appreciation was expressed for our "focus on data curation and the importance of pre-training data", etc.

As the extensive scope of this work, the reviewers raised distinct concerns from their respective perspectives:
- Reviewer PeUj questioned the necessity of pretraining a domain-specific model from scratch, given the possibility of fine-tuning existing vision foundation models (e.g., models from the DINO family). In our rebuttal, we clarified that while fine-tuning is an efficient and low-cost engineering approach, pretraining from scratch represents a scientific inquiry. It offers the potential to develop capabilities not present in general-purpose models. Specifically, a model pretrained from scratch on vast amounts of unlabeled astronomical data has the opportunity to learn intrinsic, foundational representations of the domain from the ground up, rather than merely correcting biases inherited from natural images.
- Reviewer 7FpM suggested that including results from contrastive learning pretraining would make the study more comprehensive. We incorporated the requested experimental results in our rebuttal, which was subsequently appreciated by Reviewer 7FpM.
- Reviewer zwhx inquired about the practical impact of the foundation model on the astronomy community and asked, "What makes the proposed model modifications astronomy-specific rather than general dataset tuning choices?" We addressed these questions in detail within our rebuttal.
- Reviewer PzGD focused on the relationship between our pretraining dataset (DESI-2M) and the existing Astro-76M dataset. Reviewer PzGD also requested a quantitative analysis of the impact of the additional 0.5M diversity samples on pretraining, asking why these samples improve model performance. We provided a detailed response to these points in our rebuttal.

We sincerely thank the SACs, ACs, and Reviewers for their evaluation of our paper and the constructive comments,  hoping the foregoing summary aids in your deliberation. Thank you!

Regards,

Authors of  Paper-1287

---

### Meta-Review · Area_Chair_e63U · 2025-12-17

**Summary:**

The paper benchmarks the performance of self-supervised models on astronomy data. The reviewers agree that exploring this under-explored domain is valuable to our community. However, they raise concerns about an overly broad scope and lack the in-depth study for each contribution point, which is required to appeal to the ICLR community.

The reviewers identified the following weaknesses:
- Lack of methodological novelty of C-MAE [**PeUj, zwhx**]
- Lack of details, study, and novelty on the dataset curation [**zwhx, PzGD**]
- As an astronomy benchmark, discussion on evaluation protocols and comparison with other SSL are insufficient [**PeUj, 7FpM, zwhx**]

The authors are encouraged to conduct an in-depth study on the contribution points to improve the impact on the ICLR community.
Overall, the current paper does not meet the technical depth required for ICLR.

**Reviewer Concerns:**

Resolved concerns
- Reviewer 7FpM
  - Comparison with SimSiam

Remaining concerns
- Reviewer PeUj
  - Unclear position of the paper
  - Lack of discussion on ``why it works''
  - Questionable superiority of C-MAE performance
- Reviewer zwhx
  - The scope is too broad. An in-depth study is required
  - Discussion is required to validate the contribution as a new benchmark
  - Lack of novelty in the dataset curation method
  - Lack of methodological novelty in C-MAE
- Reviewer PzGD
  - Difference of the dataset compared to uniform sampling
  - Observation of paper is not surprising due to similar reports in the medical and satellite domains

**Reviewer Scores:**

- Reviewer PeUj: Would maintain the current score. The reviewer's concerns are significant, and rebuttal comments are not enough to address them.
- Reviewer zwhx: Would maintain the current score. The reviewer's final comment indicates that the concerns remain outstanding.
- Reviewer 7FpM: Would raise the score. The rebuttal provides a comparison with SimSiam, which resolves the reviewer's concern.
- Reviewer PzGD: Would maintain the current score. The reviewer might expect to understand the novelty of the dataset curation method compared to uniform sampling. But the rebuttal comments failed to provide it.

---

### Decision · Program_Chairs · 2026-01-26

Reject